# WiFi FTM and UWB Characterization for Localization in Construction Sites

**DOI:** 10.3390/s22145373

**Published:** 2022-07-19

**Authors:** Carlos S. Álvarez-Merino, Emil J. Khatib, Hao Qiang Luo-Chen, Joel Llanes Michel, Sebastián Casalderrey-Díaz, Jesus Alonso, Raquel Barco

**Affiliations:** 1Telecommunication Research Institute (TELMA), Universidad de Málaga, E.T.S. Ingeniería de Telecomunicación, Bulevar Louis Pasteur 35, 29010 Málaga, Spain; cam@ic.uma.es (C.S.Á.-M.); hao@ic.uma.es (H.Q.L.-C.); joel7@uma.es (J.L.M.); rbarco@uma.es (R.B.); 2Innovation Department, Construcciones ACR S.A.U., 31195 Navarra, Spain; scasalderrey@acr.es (S.C.-D.); jalonso@acr.es (J.A.)

**Keywords:** WiFi FTM, UWB, position control, location fusion, indoor positioning

## Abstract

A high-precision location is becoming a necessity in the future Industry 4.0 applications that will come up in the near future. However, the construction sector remains particularly obsolete in the adoption of Industry 4.0 applications. In this work, we study the accuracy and penetration capacity of two technologies that are expected to deal with future high-precision location services, such as ultra-wide band (UWB) and WiFi fine time measurement (FTM). For this, a measurement campaign has been performed in a construction environment, where UWB and WiFi-FTM setups have been deployed. The performance of UWB and WiFi-FTM have been compared with a prior set of indoors measurements. UWB seems to provide better ranging estimation in LOS conditions but it seems cancelled by reinforcement concrete for propagation and WiFi is able to take advantage of holes in the structure to provide location services. Moreover, the impact of fusion of location technologies has been assessed to measure the potential improvements in the construction scenario.

## 1. Introduction

In recent decades, a new industrial revolution has emerged thanks to the introduction of Information and Communication Technologies (ICTs) in industrial processes [1], giving place to the Industry 4.0 paradigm. However, the construction sector remains particularly obsolete in technology adoption compared to other sectors, such as manufacturing [2,3]. The main activity of the construction sector takes place on the construction site, which is a highly changing environment, the vast majority of which is outdoors, and usually involves different actors, such as different companies and a large number of workers during the different stages of the project. Moreover, the use of heavy machinery, such as cranes or trucks, and harmful or heavy materials also come into play, which make these scenarios dangerous and whose monitoring and safety tasks are often difficult to fulfill. In this context, new ICTs are emerging that allow location and monitoring of the different resources, as well as the automation of tasks or the remote control of some elements, help to achieve a more efficient and safer construction environment [4].

In Industry 4.0, advances in the fields of robotics, AI, and Machine Learning (ML) come together to conform production to new customer demands, such as an increased customization, optimal machinery efficiency, and reduced costs [5]. Thus, the whole industry is advancing towards more flexible and adaptable scenarios through wireless environments. Wireless networks allow flexibility, scalability, and mobility that can be translated into real-world applications in the construction sector, such as remote driving [6], autonomous cranes [7], or real-time workers location and health monitoring [4].

Technological progress in recent years has focused its efforts, among other objectives, on the location of users. The Global Navigation Satellite System (GNSS) has established itself as the reference location system for outdoor navigation [8]. However, in more hostile scenarios for signal reception, such as indoor scenarios or scenarios surrounded by metallic elements, such as construction sites, it has not been successful. These types of scenarios often contain metallic objects that reflect and block signals, create multipath effects that deteriorate target location accuracy, or can create areas with coverage holes. In addition, typical construction scenarios are dynamic with constant changes due to the different phases the project goes through. Therefore, a location that meets the requirements of reliability, continuity, and accuracy for location-dependent applications, such as Augmented Reality, or Autonomous Robots, is a major challenge [9].

This paper evaluates and compares the performance of ultra-wide band (UWB), WiFi fine time measurement (FTM) and fusion of technologies in a construction scenario with an indoor scenario from a previous work [10]. This evaluation is backed by measurements taken in a real construction site, where UWB and WiFi-FTM setups were deployed. The measurement campaign included samples from several different floors in an incomplete building. The measurements are used to assess the precision of each technology individually. In addition, an algorithm [11] to opportunistically fuse the ranges obtained from different location technologies is studied. This fusion technique helps to reduce the deployment cost by reusing elements from different technologies as reference points, which may be deployed for other purposes in the construction site [10]. In locations with a high density of reference points (RPs), the system to be solved is overdetermined, i.e., it has extra information to improve its accuracy. Since all the RPs provide different accuracies, we include a weighting step that prioritises the RP that provides a better ranging accuracy [11]. In addition, the fusion technique also deals with coverage holes of certain deployments, for example, in areas where there are less than three visible RPs of a technology, fusion takes advantage of the information of other technologies to be able to offer the location service. To the best of our knowledge, there are no papers studying the precision of location systems in construction sites and that compared them with the precision of an indoor scenario. As novel ICTs emerge in the construction industry, these studies will become a necessity to properly select the location technologies for such applications.

The rest of this paper is organized as follows: Section 2 gives an overview of the different location technologies explaining the characteristics of the technologies used in this work. Section 3 explains the proposed method and the trilateration fusion algorithm. Section 4 describes the experimental setup of the scenario. In Section 5, the results obtained in the deployed scenario are presented and discussed. Finally, Section 6 presents the conclusions of this work.

## 2. Overview of Location Technologies

This section provides an overview of the two location technologies that are most commonly used indoors, and may therefore be used in construction scenarios.

### 2.1. Ultra-Wide Band (UWB)

UWB provides high distance measurement accuracy based on the Round Trip Time (RTT) protocol, even in environments with difficult propagation characteristics [12]. This technology has multiple advantages, such as centimeter-level accuracy, good obstacle penetration capability [13] and multipath mitigation in dense scenarios [14], making it indispensable for positioning in complex scenarios. UWB is also used as a wireless communication technology that supports high throughput due to the use of a very wide spectrum. UWB uses very short time pulses of a few nanoseconds that occupy a wide bandwidth. UWB signals are centered on 6.5 GHz with a bandwidth greater than 500 MHz. The latest market trends show that UWB may soon become a de-facto standard (albeit this prediction is recently being challenged by IEEE 802.11 mc). Therefore, some smartphones have already integrated UWB chips in recent years. The main drawback is that, in order to achieve a short pulse width, the UWB device has a high power consumption for a single packet transmission [15,16]. Thus, using the RTT protocol, which needs the exchange of multiple packets, will increment the energy consumption.

### 2.2. WiFi Fine Time Measurement

The IEEE 802.11mc standard includes precise fine time measurement (FTM) for distance estimation to the router by time stamping using the RTT protocol [17,18]. This version will transform the indoor positioning industry in the coming years as WiFi infrastructure and connectivity is widely adopted. The protocol accurately estimates the distance of any user supporting the WiFi FTM protocol without the need to be connected to the router [19]. The information is calculated on the device to preserve privacy, as sensitive location information is not shared between network nodes. In [18,20], the accuracy for WiFi FTM positioning is estimated to be around one meter in real-world scenarios with dense deployments of WiFi routers or access points (APs).

## 3. Location Computation

Although fingerprinting claims to provide high accuracy with low infrastructure deployment, it has some drawbacks that make it unfeasible for the construction case. First, it requires complex training that makes it impractical to cover the entire infrastructure. Secondly, the periodicity of this training becomes very frequent due to constant changes in the environment. Thus, the trilateration method seeks to find the final position of the user through the intersection between geometric shapes, such as circles or hyperbolas defined by the distance between the target and the different RPs [14,21,22].

Although the received power may not follow a specific propagation model, if the environment does not change drastically, it tends to remain static over time.

Since the distances contain errors in the measurements due to different factors, such as reflections or blockages, these distances do not intersect at a point but generate an area of uncertainty which is where the solution to the problem lies. Therefore, the Weighted Least Square (WLS) iterative method finds the optimal solution to this problem as follows:(1)$$\begin{array}{c} A=\left(\begin{array}{cccc} \frac{x^{n}-bs_{x_{0}}}{\rho_{0}} & \frac{x^{n}-bs_{x_{1}}}{\rho_{1}} & \cdots & \frac{x^{n}-bs_{x_{n}}}{\rho_{n}}\\ \frac{y^{n}-bs_{y_{0}}}{\rho_{0}} & \frac{y^{n}-bs_{y_{1}}}{\rho_{1}} & \cdots & \frac{y^{n}-bs_{y_{n}}}{\rho_{n}} \end{array}\right)\\ y=p^{n}-p\\ \Delta p={\left(A^{⊤}WA\right)}^{-1}A^{⊤}Wy\\ p=p^{n}+\Delta p \end{array}$$
where *A* is the Euclidean distance matrix of the computed position ($p^{n}$) which is defined as p(x,y) in the *n* iteration, $\rho_{i}$ is the pseudodistance from the target to the *i* reference point and $bs_{i}\left(x,y\right)$ is the coordinate of the reference point in the second dimension. *W* is the weighted matrix and (${\mathrm{bs}}_{i}$) are the coordinates of the different RPs. The innovation vector y computes the difference between the estimated and the initial position p which is updated until the variation Δp does not exceed an arbitrary threshold.

In trilateration, it is usually assumed that the distance measurement information comes from a single technology. However, in [11] a scheme for merging ranges from different technologies is presented. Moreover, in [10] the UWB and WiFi FTM technologies are presented in a real indoors scenario where the result of locating users using these two technologies separately and in fusion is shown. In the present work, the same algorithm is presented to compare the performance of the different technologies in different scenarios. The use of fusion in trilateration improves the accuracy of the final estimated location. In addition, fusion in trilateration also provides seamless navigation between areas served by different technologies (e.g., outdoors where GNSS can be used, and indoors with WiFi deployments, using other distance measurement technologies to cover missing ranges at the borders). Fusion leverage signals from isolated high-accuracy landmarks or that are part of incomplete deployments, such as in situations where a dense deployment is not possible, such as in stages of a construction where the addition of walls has caused blockages, the removal of scaffolding has reduced the mounting points for RPs or even in occasions where part of the infrastructure has been destroyed (e.g., fires, earthquakes, etc.). In these scenarios, fusion can compensate for missing RPs with portable APs to provide high-precision location.

The classical Least-Square algorithm is highly influenced by outliers. However, the Maximum Likelihood Estimator (MLE) estimator evaluates signal accuracies to enhance the location service. MLE obtains the parameter $\hat{\theta }$ which determines a probability density function p(X=x|θ) of a continuous variable based on $x_{1},x_{2};\ldots ,x_{n}$ which are independent observations of the distribution [23]. In this work, MLE weights the ranging information obtained by different RPs depending on the error of the ranges compared with the final solution to insert this information in the WLS with the *W* matrix. The system stores the error of each RP iteratively with a temporal window and weights the sources according to their standard deviation. Supposing that $X=\{X_{1},X_{2},\ldots ,X_{n}\}$ with distribution $F_{\theta }\;\mathrm{being}\;\theta =\left\{\theta_{1},\theta_{2},\ldots ,\theta_{n}\right\}$ that follows the density function $f_{\theta }\left(x\right)$ [24]. So, the likelihood function of the observation is given by:(2)$$L\left(\theta ;X\right)=\prod_{i=1}^{n}f_{\theta }\left(X_{i}\right)$$

The MLE estimates the best candidate that optimally maximizes *L* as seen below:(3)$$\hat{\theta }=argmax\left(log\left(L(\theta ;X)\right)\right)$$

Assuming that observations follow a Gaussian distribution [18,25,26], the estimator calculates the parameters of mean and standard deviation that best suits Equation (Equation 3). Thus, the given observation vector to the MLE follows a normal distribution, as indicated in [8], which provides the optimal value. In this work, MLE obtains the weights of the different RPs based on the error from the last *N* time epochs which may improve over-determined location systems.

## 4. Experimental Environment

In this section, an experimental setup for evaluating UWB, WiFi-FTM, and fusion in a real construction scenario is described. To show the performance of the different technologies with real data, a setup of UWB anchors and a WiFi FTM APs has been deployed. In this work, two experiments have been performed to evaluate the ranging and location accuracy in the same floor of UWB, WiFi and fusion, and to evaluate the penetration capacity and accuracy of UWB and WiFi in different floors (one above and below from where the deployment is set-up). The first experiment takes place in the basement floor −1 during the construction phase of a building site, as can be seen in Figure 1. In this floor, the view of the sky is completely cancelled, making unfeasible the use of GNSS. The construction site was at a stage where the structure of the floors was built although the walls had not yet been built as shown in Figure 1a. Figure 1b shows the steel bars inside the reinforcement concrete that act as Faraday cage for the signal propagation. This structure is present in the walls, floors, ceiling, and columns. Thus, the second experiment evaluates the penetration capabilities of UWB and WiFi. Figure 1c shows how the anchors are attached to the walls for the measurement campaign. Duct tape was used in order to avoid interfering with the construction works, and taking into account that it does not affect the propagation of wireless signals. This deployment strategy was also drawn in conjunction with the construction company, which provided guidance on typical practices and material limitations of the environment. Specifically, this approach was recommended due to its low cost, low intrusiveness, and high flexibility.

In the first experiment, the distribution of the UWB anchors and WiFi APs are represented in Figure 2 as blue and red triangles and indicated as UWB or WiFi *X*, respectively. The yellow dots represent the ground truth of the path there and back where the measurements are taken. In this experiment, all the measurements were taken in LOS conditions and for UWB4 and UWB6 in partially NLOS conditions. Two path there and back measurement recollections were performed to have a sufficient dataset, over 150 samples that were collected statically during each 30s measurement. UWB anchors are set at different heights (indicated in Figure 2) and the WiFi APs are placed on the floor to avoid falls. In the second experiment, the green boxes (1, 2, and 3) show the positions used to measure the penetration capabilities of UWB and WiFi one floor above and below of each point. In this experiment, we evaluate the penetration capabilities, i.e., the number of packets that can reach the UE, and the ranging accuracy of how it degrades from LOS to a NLOS scenario. In the positions marked by the green boxes, the UWB and WiFi devices are placed together on the floor (height = 0m). The measurement campaign at each point was of 5 min with an update date of 3 Hz.

A Google Pixel 3 acts as the location target. This smartphone runs Android 9.0 and supports WiFi FTM RTT. An application has been programmed to collect all distance measurement data from the RPs seen by the terminal: anchors for UWB and APs for WiFi. However, Google Pixel 3 does not support UWB technology yet. Thus, an UWB device is attached to the smartphone (acting as a tag, i.e., location target) and connected via Bluetooth Low Energy (BLE) to the smartphone which reads the UWB data. The developed application relays the captured data via WiFi to a server where it is stored and processed. The database is a Flask server with a MySQL database that is configured on a Windows 10 laptop. Samples have been collected in an offline phase to check the accuracy of the positioning results with a sampling rate of 0.3 Hz.

The UWB devices—anchors (reference points) and tag (location target)—are based on Decawave DWM1001 devices which compute the range estimation via RTT protocol [27]. The UWB devices transmit with a power of −14.3 dBm and they are centered in 6.5 GHz [27]. One limitation in the performance of these UWB devices is that the tag can only receive information from four anchors simultaneously due to the default firmware that DWM1001 devices have installed [13]. Thus, despite of the high density of UWB anchors, the positioning algorithm with UWB only will use up to four anchors that does not exploit the full environment information. The WiFi APs are Google WiFi routers which are configured to work at 5GHz to support the WiFi FTM RTT protocol [28]. Additionally, different bandwidth gives different precision as indicated in [29]. For ranging estimations at 90% CDF error, it is expected to have the following tolerances: 80 MHz (2 m), 40 MHz (4 m), and 20 MHz (8 m).

## 5. Results

This section describes the results obtained in each experiment to demonstrate the performance of the different technologies for positioning accuracy and penetration capacity.

### 5.1. Accuracy of UWB and WiFi in the Same Floor

Figure 3 shows the cumulative density function of the horizontal error (x–y axes) as a result of trilateration obtained with UWB, WiFi, and fusion using trilateration. It also compares the performance with the indoor scenario (Case 1) presented in [10].

Table 1 displays the relevant statistical parameters of the first location experiment, i.e., the mean, standard deviation of the error, and the 2σ parameter (95% of the sorted error) compared with the ground truth in UWB, WiFi, and fusion in both scenarios, construction site and laboratory, as a result of the trilateration with MLE algorithm.

In addition, Figure 4 and Figure 5 represent the ECDF of the ranging accuracy of the different UWB and WiFi devices to the target during the measurement campaign in LOS and partially in NLOS (mainly for UWB4 and UWB6) conditions.

As can be observed, UWB seems to perform better than WiFi for ranging accuracy and, therefore, in location estimation. In general terms, the geometry of the deployment and features of the environment (e.g., construction site with LOS/NLOS, reinforced concrete walls and floors, etc.) are key factors that may change the performance of a location system. Indoor scenario ranging outperforms compared to the construction site, and the fusion algorithm enhances both UWB and WiFi performance in both scenarios. Despite having all UWB anchors and WiFi APs located in areas with good propagation conditions, the 90% percentile of the positioning error is above a meter in all construction cases due to multipath effects, geometry distribution, which leads to dilute the final precision, that affects both UWB and WiFi FTM although, for WiFi, it can be seen that the effects are slightly smaller. Despite having worse ranging performance in WiFi, the final solution is similar to the UWB final results. As it can be seen, more ranging information in fusion improves the geometry of the system and overdetermines the LS algorithm, which results in a better performance. In this case, the fusion algorithm benefits from the data of estimated ranges to RPs obtained from multiple technologies (normally 7 ranging data), when UWB only captures normally 4 ranging data and WiFi a maximum of 3 ranging data. The full potential of fusion is realized in cases where the scenario is such that RPs of a single technology do not offer full coverage. In other words, in scenarios where points where less than 3 RPs of a single technology are visible. In these cases, fusion may complement the missing RPs with a different technology. Nevertheless, in the setup of this experiment, our objective was not to demonstrate this opportunistic nature of fusion.

### 5.2. Penetration Capacity and Accuracy of UWB and WiFi in Another Floor

In this second experiment, we measured the location provided by the RPs one floor above and below the scenario. The first observation was that no signal was received from UWB. In other words, to have location in a floor, the UWB anchors must be installed in the same floor. Figure 6 shows the percentage of the packet loss of the RTT packets at the different points (1, 2, and 3) on the lower (Floor −2) and upper (Floor 0) floors of WiFi. These loss rates reflect a reliability that could be sufficient for non-critical applications; for instance, worker tracking or tool location which normally require update rates of a few Hz [30]. Half of the measurement points (Below 1 at Floor −2, Above and Below 3 at Floors 0 and −2, respectively) show a much higher loss rate due to the reinforcement concrete which block the signal. In one of the measurement points (Above 3 at Floor 0), no packets were recorded at all. In contrast, in the other half of the measurements (Above 1 at Floor 0, Above and Below 2 at Floors 0 and −2, respectively), the RPs can communicate with the smartphone due to some holes present in the structure of the building among different floors that are shown in Figure 7a–c.

Figure 8 shows the ranging error (i.e., the error in the distance measured to the WiFi-FTM RP) for each of the measured points. It can be seen that the Above 2 at Floor 0 and Below 3 at Floor −2 are the most precise points, despite having high packet losses. In the points with lower packet losses, precision is slightly lower and there is a higher tendency for outliers. This is due to the impact of the holes in the building structure; while they help propagation, they also introduce a higher error due to multipath. The error maintains below 5 m of ranging error, despite the fact that in some cases the packet rate loss exceed the 80% of the transmitted packets. This means that in cases where only signal penetration is available for positioning, the ranging error maintains stability in general.

As it can be observed, the error estimation reasonably increments compared with performance of the WiFi ranging information in LOS conditions from the first experiment in Figure 5.

## 6. Conclusions

This paper presents the results of a measurement campaign of UWB and WiFi FTM in a construction scenario for location purposes. The goal of the experiments is to compare the accuracy, coverage, and penetration capability of UWB and WiFi technologies and the fusion of technologies in this type of dynamic scenarios. UWB has demonstrated to provide better ranging accuracy, however, WiFi has demonstrated robustness against blocks in the scenarios with better propagation performance and penetration capabilities. The measurements show the elements of the construction site affect in UWB and WiFi ranging estimation compared with an indoor scenario. Moreover, it can be observed that fusion can improve the accuracy of location in all scenarios.

Penetration measurements show that reinforcement concrete completely cancels UWB propagation and WiFi is able to benefit from holes in the structure to achieve location. However, in cases where no holes are present, WiFi performs with difficulties for positioning, but still manages to report ranges.

## Figures and Tables

**Figure 1 sensors-22-05373-f001:**
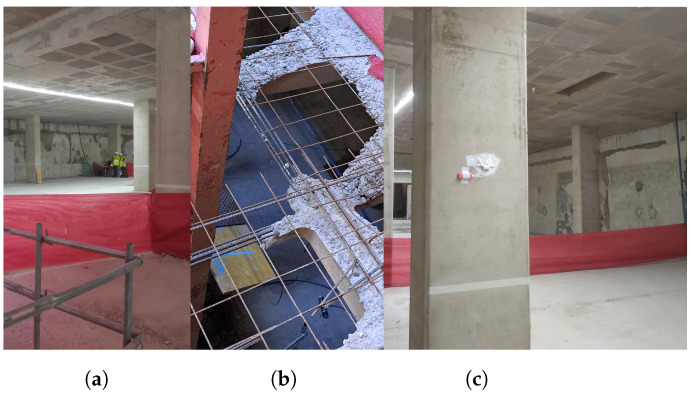
Pictures of the scenario: general view (**a**), structure of the floor and ceiling (**b**) and setup of an UWB anchor (**c**).

**Figure 2 sensors-22-05373-f002:**
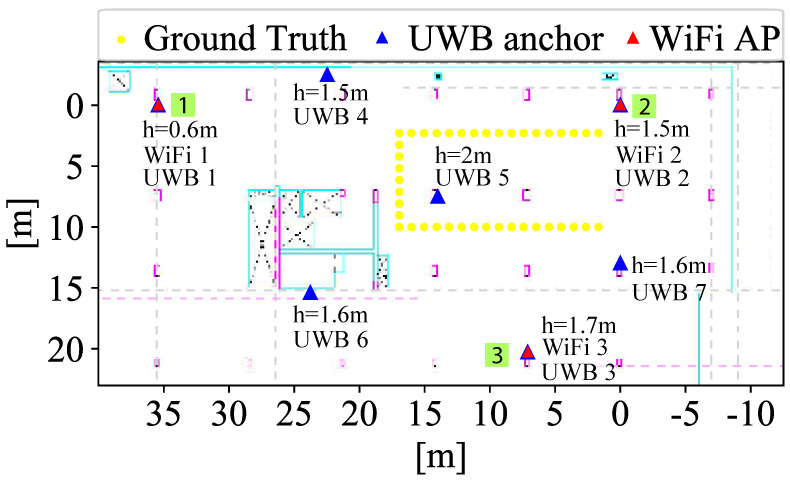
Scenario with UWB and WiFi technologies.

**Figure 3 sensors-22-05373-f003:**
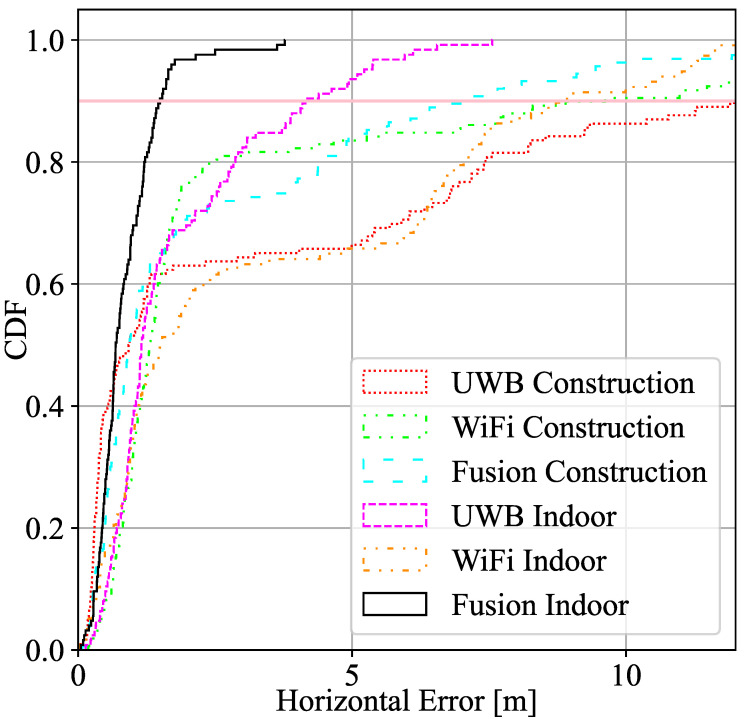
CDF Horizontal error distribution of UWB, WiFi FTM, and fusion in the construction site and in a indoor scenario.

**Figure 4 sensors-22-05373-f004:**
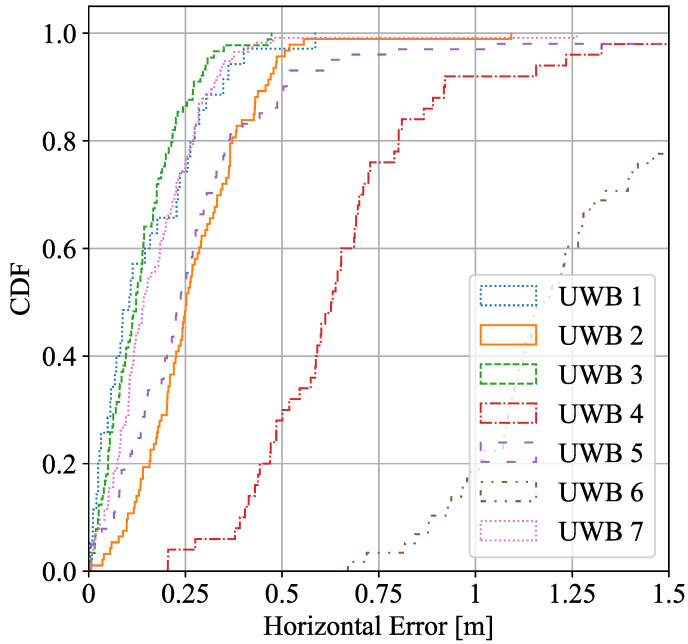
UWB ranging estimation error in LOS conditions.

**Figure 5 sensors-22-05373-f005:**
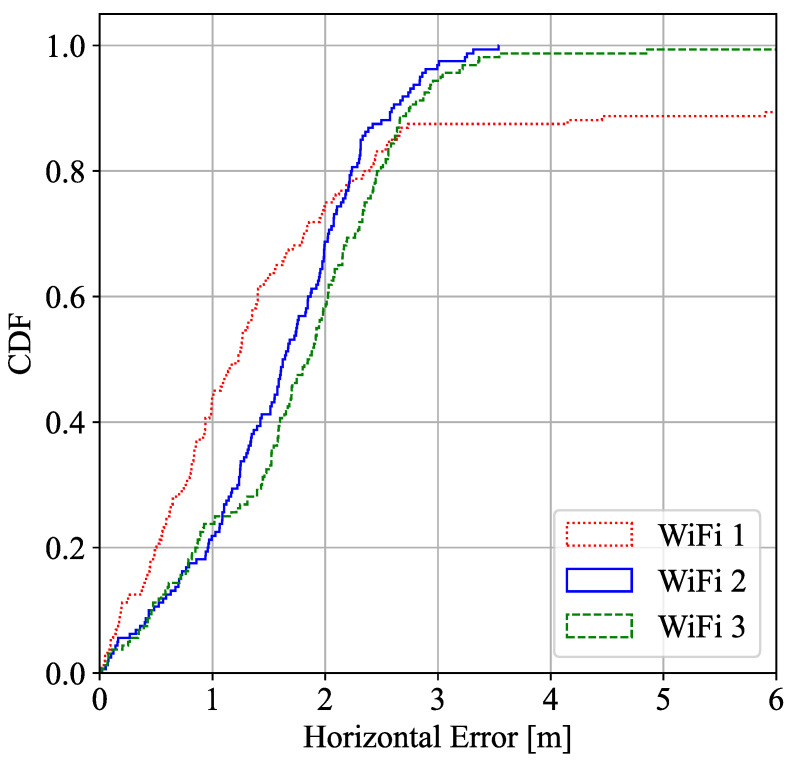
WiFi ranging estimation error in LOS conditions.

**Figure 6 sensors-22-05373-f006:**
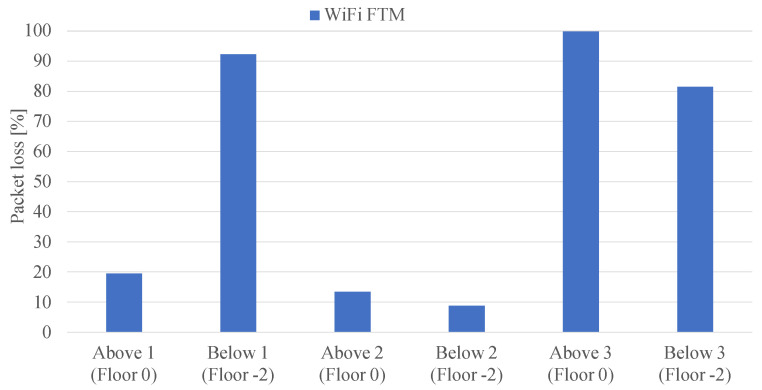
Percentage of WiFi coverage on the floor above and below the measurements taken.

**Figure 7 sensors-22-05373-f007:**
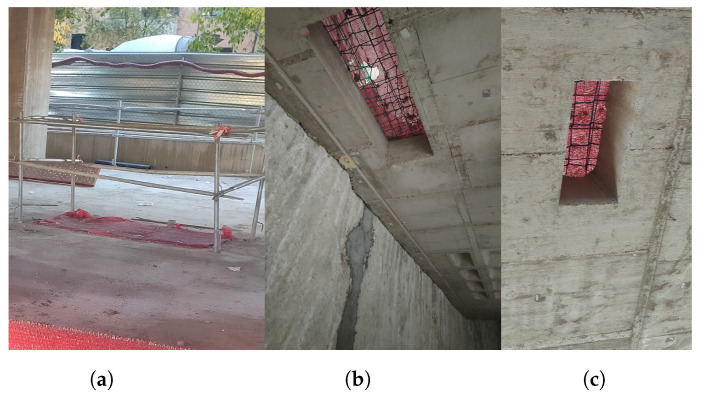
Images of the connection between floors.

**Figure 8 sensors-22-05373-f008:**
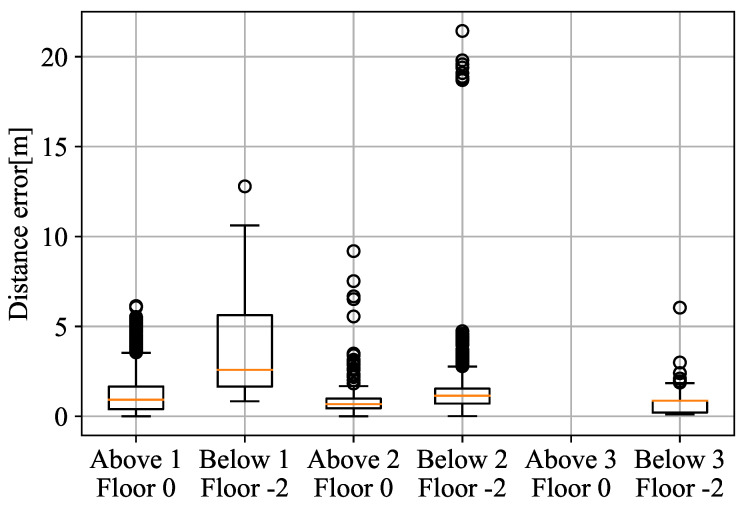
WiFi error on the floor above and below measurement.

**Table 1 sensors-22-05373-t001:** Comparison of the horizontal error between UWB, WiFi FTM, and fusion in the construction site and indoor.

	Mean [m]	Standard Deviation [m]	90% of Error [m]
**UWB Construction**	3.69	4.70	13.51
UWB Indoor	1.82	1.54	5.22
**WiFi Construction**	3.02	4.45	14.14
WiFi Indoor	3.53	3.55	11.11
**Fusion Construction**	2.40	3.05	9.41
Fusion Indoor	0.86	0.58	1.65

## Data Availability

Not applicable.

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
