# Peer review of "WiFi FTM and UWB Characterization for Localization in Construction Sites"

_sensors, 2022, doi:10.3390/s22145373_

Round 1

Reviewer 1 Report

I have some doubts and comments about the work presented:

- In the abstract, line 5, is missing a space after the period.
- In section 2.1, the UWB signals do not have to be centered at 3.5GHz. In fact, in the DWM1001 chip referenced in 9, the frequency band is centered at 6.5GHz. This module is based on the DW1000, which allowed different frequency bands to be selected between approximately 3.5GHz and 6.5GHz.
- The claim that a UWB device has a high power consumption needs some kind of explanation. High compared to what? Compared to Wifi?
- In section 3, is there any proof or study that the distance estimates obtained by WiFi or UWB follow a normal distribution? Because without this certainty, the distribution that the MLE will give you will be far from the optimal value.
- In section 4 the figures are referenced as 1 a, 1 b and 1 c when in the caption they appear as (1), (2) and (3).
- In section 4 the hardware description should appear at the beginning of the section.
- I don't quite understand what data was captured in the experiment. Distance measurements? position estimates? Text says that a limitation of the DWM1000 (DWM1000 or DWM1001? the name in the text is different than the reference) is that it can only receive data from 4 anchors "simultaneously". As far as I know these devices have only one antenna, so simultaneously only can communicate with another device. On the other hand, the "4 anchor"s number only appears in the datasheet referenced as a limitation if the internal positioning algorithm of the DWM1001 is used, but it does not say anything about any limit if the module is used to make sequential ranging measurements (which is something that in fact can be done).

- The placement of the UWB anchors glued to a concrete wall causes them to suffer much more interference than if they are placed at some distance.

- On the other hand, in experiment 1, when talking about 150 samples, are they ground truth points or measurements? Were the measurements taken in motion or static at each point? If they were static at each point, for how long?

- What is called Horizontal Error in figure 3 refers to the error in the x-y axes?

- When the results of experiment 2 are shown, we talk about ranging, but there is no ranging data averaged in the same plant, i.e., we only have NLOS data but not from a more or less LOS environment. Because seeing the ranging accuracy in NLOS, in LOS should be obtained a very high accuracy.

-The number of references is very low, and many of them are references to previous work by the authors. Other proposals in the literature should be analyzed, and there are some.

Author Response

Here is the review of the article for the first reviewer

Reviewer 2 Report

"WiFi FTM and UWB characterization for localization in construction sites"

(I am happy that this is not another indoor localization paper about RSSI, fingerprinting and/or "machine learning")

The two technologies explored here for indoor localization (UWB and FTM RTT) appear to be at the heart of an inter-company rivalry: Apple is backing UWB and Google is backing FTM RTT. So an unbiased assessment of these technologies in practice would be of great interest to the reader. There is much information (and misinformation) on this topic available online.

FTM RTT was first standardized by the Wi-Fi Alliance in 2016 and has been supported in Android since at least 2018. Apple U-1, which supports UWB in the 6-8.5 GHz range came out in the 2019-2020 period.

The reader most likely would love to see a clear answer about which technology is better, but this paper, with its limited experimentation, does not answer that question. We'll just have to wait for the technologies to battle it out.

Section 2.1:

The main issue with UWB is that it is limited to very low power because of the potential interference with many channels of communication covered by the wide bandwidth.

    "The FCC power spectral density emission limit for UWB transmitters is -41.3 dBm/MHz. However, the emission limit for UWB emitters may be significantly lower (as low as −75 dBm/MHz) in some segments of the spectrum.

This limits UWB's useful range and prevents penetration of many materials, as the authors discovered.

    "There is one drawback which is its short range"

In section 2.1:

"UWB signals are centered on 3.5 GHz". " Later in section 4, "they are centered in 6 GHz" Which is it?

The statement "market trends show that UWB will soon become a de-facto standard for positioning" is unsupported. Apple may hope that this was the case, but it is not clear whether UWB's short range will limit its application scenarios. (There are also issues about privacy - which do not arise with IEEE 802.11mc). For the last few years, FTM RTT has been available on some standard smartphones and some Wi-Fi routers. The same cannot be said of UWB - so far. It seems to require extra equipment that needs to be patched into the phone, as done here, and often also requires remote processing, with obvious privacy and throughput and scaling concerns.

Why could the tests not be performed with a phone that supports UWB if it is soon to become a de-facto standard?

In section 2.2:

"In [ 12 ], the accuracy for WiFi FTM positioning is estimated to be around one meter in real-world scenarios with dense deployments of WiFi routers or access points (APs)"

This is overly optimistic:

(1) Some papers on FTM RTT talk about (maybe) 1-2 meters in open spaces (like an open atrium). e.g. “How To Achieve 1 Meter Accuracy In Android” by Frank van Diggelen, Roy Want, and Wei Wang. But in a building, accuracies tend to be considerably lower because of interactions with materials in the building.

(2) Is this the raw measurement accuracy of individual FTM RTT measurements or is it the accuracy of location after "multi-lateration"? That is, there is some "dilution of accuracy" as in GPS.

(3) Is this the st. dev. reported by the API, or is it the actual st dev as obtained from repeated measurement? The latter being considerably larger because of effects of the environment such as interference and fading (the phone reports the st. dev. of measurements taken in rapid succession and thus is not picking up these changes).

Also, I see no mention of the offsets in FTM RTT. That is, while (in open space) RTT linearly increases with distance, there is an offset that has to be subtracted out. This can be several meters. Although for Google Pixel and Google WiFi it tends to be small, perhaps 1 meter. Obviously, if all measurements are offset by this bias the resulting location is not as good as if they were taken care of.

Section 3:

It should be made clear that "trilateration" and "multilateration" are just one approach to estimating position and actually not considered the best options (Even in this paper the authors resort to some probabilistic method using a history of measurements - end of section 3).

Then, the notation in equation (1) needs to be explained and refined. The double bar notation usually indicates the norm of a vector (or matrix) resulting in a scalar, but A is a matrix? Then, if this is the norm of the difference between measured and actual distance, indexed by i, we still only have a vector, not a matrix. What is the other dimension of the matrix? Are we stacking up difference vectors (not norms of difference vectors)? Is the matrix 2 x N, where N is the number of measurements?

What is p-hat? Is it the result of the previous iteration? That is, p^+ will become p-hat on the next iteration? Maybe use superscripts to indicate this? Use p^{(n)} for p-hat and p^{(n+1)} for p^+?

The last part of section 3 mentions the sensitivity of least squares methods to outliers and then introduces an MLE estimator. It is not clear whether this involves keeping a history, whether the "the system stores the error of each RP iteratively with a temporal window". Is this somehow using a Bayesian grid update or particle filter (as in some reported FTM RTT systems)? Needs a bit more detail in the explanation. Also, it is unclear whether this or the "multilateration" is used in the experiments.

Then "Assuming that observations follow a Gaussian distribution, ..." is not a good assumption for FTM RTT or UWB measurements, which have offsets, asymmetric probability densities, and outliers. It also contradicts the premise of this section, which is that one has to deal with outliers. So, is this aimed at a Kalman filter implementation (which in its basic form requires Gaussian independent noise assumptions)?

Section 4:

Explain what a UWB anchor is and what a UWB tag is, please.

"Over 150 samples" This is rather meager when other indoor localization papers mentioned thousands or even tens of thousands of samples.

"WiFi APs are placed on the floor to avoid falls". OK, perhaps in an empty building this may make sense. In an occupied building the best place is above the heads of people. On the floor would NOT be a good deployment since the signals would pass through many objects on the way to and from the APs and the phones.

"Samples have been collected in an offline phase to check the accuracy of the positioning results with a sampling rate of 0.3 Hz."

That is rather a slow update rate. A person will move 4 to 5 meters between samples. Is it limited by the fact that a remote server is involved? Will the phone in the future be able to do this computation without a distant server? Is it possible to do this in real time as required by most applications?

"The UWB devices transmit with a power of -14.3 dBm" That seems about right, given government agency limit of -41.3 dBm/MHz and 500 MHz bandwidth. But that is only 37 micro-watt! This is the price paid for the greater accuracy in distance measurement.

"The WiFi APs are Google WiFi routers which are configured to work at 2.4GHz. To the best of our knowledge, these routers do not use the 5GHz for FTM."

This can't be right. First of all, the Google Wi-Fi boxes work in both 2,4 GHz and 5 GHz. Secondly, they do not support FTM RTT in the 2,4 GHz band! Then, suppose they somehow did really operate in the 2,4 GHz band, note that the accuracy of FTM RTT is inversely proportional to bandwidth and that the 5 GHz band allows for 80 MHz bandwidth, while the 2,4 GHz band is restricted to 40 MHz. So the accuracy would be about twice as good in the 5 GHz band.

Section 5:

In Figure 3, please expand the horizontal scale by a factor of 2 (and reduce the range of horizontal error to end at about 12 meters instead of about 25 meters). The reason is that all the interesting part of the figure is squeezed into the leftmost block (0 to 5 meter) and hard to distinguish.

Also, typo in "Contruction" (three times)

Just to be clear, what is called "horizontal error" here is an error in the result of "trilateration" not the raw st dev reported by the range measurement? Clarify, please.

"As it can be observed, the scenario highly influence on the technology-ranging202 performance" This sentence isn't easy to parse. Do you mean that the results in an occupied office building are different from those in an empty concrete shell of a building? Please explain.

Table 2: please indicate the units (meters presumably?)

In figure 4, remove "UWB" from the figure and remove it from the caption, since UWB did not penetrate, and no part of the figure shows anything to do with UWB.

"...the RPs can communicate with the smartphone due to some holes present in the structure of the building among different floors that are shown in Figure 5."

How sure are you of that? It may be that the RF signal detected passed straight through the concrete. This can be tested by covering the hole with a metal plate. I would love to know the result of such an experiment. How thick is the concrete floor? What evidence is there for reflecting surfaces to complete a multipath circuit? What is the distance measured by FTM RTT in this case? Is it considerably larger than the actual distance? In Fig. 6, are those positive distance errors only?

Overall:

There is little reference to existing papers on FTM RTT (or UWB), including many papers published in MDPI Sensors. Please add some more to avoid the impression that this is a brand-new field.

Author Response

Here we present the review for the second reviewer

Round 2

Reviewer 1 Report

In general I think the work has improved a lot after the last changes. But I still have some questions that I would like to comment on. First in relation to the answers of the first review:

Comment 3:

This is not the case. The power consumption depends on both the implementation and the operating mode of each device. In Decawave APS001 APPLICATION NOTE you can see an explanation of all the factors involved in the DW1000 power consumption calculation, depending on the selected operating mode. A UWB tag pinging in TDOA mode does not need to exchange TWR messages, so its power consumption is limited to that needed to transmit a pulse periodically. Reference 3 does not justify the statement raised, since two UWB devices are compared, while reference 7 says that the UWB device used uses 140 mA in TX while the ESP32 wifi module consumes 190 mA. And I repeat that using a UWB tag in TDoA mode, you only need to emit the pulses, not receive them. And the consumption on the tag (which is the portable device by definition) would not be then "the main drawback" of the technology, especially if compared to wifi. I would rephrase this sentence to narrow the scope of the statement or qualify it enough so that it can be considered correct in all contexts.

Comment 6.

Hacking is not necessary, but development is. I would then make it clear in the text that the limitation of ranging to only 4 anchors per sequence is a limitation of the firmware used, not of the technology, and could be overcome with a software modification.

Comment 7

This is a limitation that is important for analyzing the final results, so it should be clarified in the text. Something similar to what appears in this answer.

And a couple of comments on the latest version of the text:

- On line 125, the sentence starting with "In this case, the same algorithm...." would be better changed to something like "In the present work...", otherwise it may seem that "this case" refers to the previous sentence.

- In line 285 the sentence "The limitations on the density of deployments (e.g. limitation of 4 simultaneous RPs in UWB) may limit the precision of location systems in construction." I think it does not contribute anything, since the limitation in the number of sensors will affect the precision of the system in any scenario, not only in construction. And again the limitation of 4 nodes in UWB is a mere software issue, not a hardware limitation, much less a technology limitation.

Author Response

Comments are responded in the PDF.

Reviewer 2 Report

Thank you for the revisions and responses to the reviewer's comments. Seems ready to go now.

I do have some remaining comments you may find interesting.

re: comment 7 (bias/offset correction)

"Moreover, in the procurement of the bias is similar to fingerprinting approach. In case of changes in the environment, the bias would have changed"

No, this is a one-time, simple measurement. It remains fixed for a particular model of AP (although it may be different for different bands, such as 2.4 GHz and 5 GHz). It is not at all like fingerprinting, which has to be redone when things change. For some access points, this is important since the bias can be as large as +/- 6 to 8 meters. For Google Pixel <-> Google WiFi it is less than a meter, so maybe can be ignored.

re: comment 8 (methods other than "multi-lateration")

I was referring to more sophisticated methods for RTT such as various "filtering" approaches such as particle filter, Kalman filter, Bayesian filter etc. These work better than simple geometric/algebraic "multi-lateration" (Particularly the discredited "linearized" version --- which is still foolishly referred to).

"Fingerprinting claims to be much better that multi-lateration."

What? Fingerprinting using RSSI has been a long struggle leading to hundreds of papers, but no real-world deployment. The idea of RTT is to come up with a method that actually measures something tied more closely to distance. Again, there appears to be a misunderstanding. I was thinking of more sophisticated localization methods than "multi-lateration" - not going back to the dark ages of RSSI :-)

re: [3] Wi-fi rtt (ieee 802.11mc). https://source.android.com/devices/tech/connect/wifi-rtt. Accessed: 01/07/2022.

This is a web page useful to Wi-Fi hardware vendors for a low-level interface to Android HAL. Not for users of FTM RTT.
A better reference may be
https://developer.android.com/guide/topics/connectivity/wifi-rtt

re: [30] Google Support. https://support.google.com/googlenest/thread/44036979/google-wifi-transmit-power-value?hl=en. Ac-
cessed: 28/09/2021

The internet is ephemeral. This link already is "stale". I don't have a good replacement, since Google seems very keen not to let you know what power levels are on their WiFi box. (Perhaps because of wackos that worry about "mind control" via 5 GHz WiFi :-)) Wikipedia gives upper limits for various countries, but as far as I know, Google WiFi is well below those upper limits https://en.wikipedia.org/wiki/List_of_WLAN_channels#Standard_power

By the way, for my elucidation, I was hoping you would make some RTT measurements from one side of a large concrete obstruction (like a wall or floor) with no holes and estimate how much larger the average RTT distance reported is compared to what it would have been without the concrete. This is a hard measurement to make well unless you have access to a building site. We find that with 0,3 m of concrete we see an additional meter or so distance due to the concrete (in addition to the 0,3 m). But it is hard to make this an accurate measurement when the obstacle is finite in size and signal can diffract around its outer edges.

Author Response

Comments are responded in the PDF.
